# Study on the Characteristics of Vacuum-Bagged Fermentation of Apo Pickle and Visualization Array Analysis of the Fermentation Process

**DOI:** 10.3390/foods12193573

**Published:** 2023-09-26

**Authors:** Jiawei Liu, Mengyao Wang, Ying Huang, Hai Sun, Haiying Liu

**Affiliations:** 1School of Food Science and Technology, Jiangnan University, Wuxi 214000, China; 6210112050@stu.jiangnan.edu.cn (J.L.); 6200112076@stu.jiangnan.edu.cn (M.W.); 6220111044@stu.jiangnan.edu.cn (Y.H.); 2Jiang Xiao Yao Food Technology Co., Ltd., Suzhou 215000, China; sh@jxyjc.com; 3State Key Laboratory of Food Science and Technology, Jiangnan University, Wuxi 214000, China

**Keywords:** apo pickle, fermentation, flavor, microbial diversity, colorimetric sensor array

## Abstract

Apo pickle is a fermented food with a long edible history in the Jiangnan region of China. Traditionally, plastic bottles are used as Apo pickle’s fermentation containers, and artificial bottling costs are high. The goal of this study is to compare the fermentation effects of Apo pickle fermented under low pressure in a vacuum bag (VBA) and Apo pickle fermented under normal pressure in plastic bottles (TBA) to determine the feasibility of fermenting Apo pickle in a vacuum bag rather than a plastic bottle, thereby lowering production costs. At the same time, a gas-sensitive colorimetric sensor array (CSA) was developed to distinguish different fermentation stages of Apo pickle. The results revealed that the main genera in the initial and final phases of Apo pickle fermentation were *Weissella* and *Lactobacillus,* unaffected by fermentation containers. At the same fermentation time, the abundance of *Lactobacillus* and the content of flavor substances in VBA were higher, and the fermentation speed of VBA was faster at 0–15 d, so a vacuum bag could be used instead of a plastic bottle. The CSA could discriminate between different fermentation procedures of Apo pickles with an accuracy rate of 93.8%. Its principle is similar to that of an electronic nose. It has the advantages of convenience, rapidity, and no need for professional equipment, so it can be used as a new method to judge the fermentation degree of apo pickle.

## 1. Introduction

Apo pickle is a traditional pickle in the Jiangnan region of China. It is made of *Brassica napus* by pretreatment (washing and drying), salting, dehydration, and sealed fermentation. Under anaerobic conditions, *B. napus* can be fermented and matured through microbial metabolism, salt anti-corrosion effects, and a series of biochemical reactions to obtain a unique flavor, which local people enjoy. The flavor of pickled vegetables is a major aspect influencing their quality [1], and the flavor is influenced by microorganisms, such as lactic acid bacteria (LAB), molds, and yeasts in the fermenting environment. LAB is a key player in fermentation [2]. However, fermented vegetables’ types and dominant LAB strains differ in different countries and regions. For example, the dominant strains in the early stage of suan-cai in Northeast China are *Leuconostoc* sp., *Lactobacillus taiwanensis*, and *Lactobacillus acidophilus*, and the dominant strain in the later stage is *Lactobacillus coryniformis* [3]; *Weissella confusa*, *Leuconostoc citreum,* and *Lactobacillus* sake are the main LAB in kimchi [4], and the main LAB in fermented Spanish olives are *Lactobacillus* (81.94%) and *Leuconostoc* (10.42%) [5]. 

Apo pickle is fermented by a method of sealing after draining the water. It is difficult to control the temperature if large containers are used for Apo pickle fermentation, resulting in uncontrollable product quality and safety problems. Therefore, small-capacity containers, such as small plastic bottles, are often used for fermentation in production. Because the automatic bottling technology of *B. napus* has not been realized, a great deal of manpower and time are needed for bottling every production run, which greatly increases the cost. Because LAB plays a leading role in vegetable fermentation, most of the LAB play a role under anaerobic conditions. In this study, the method of vacuum bagging was considered to provide a low-oxygen environment for pickles fermentation to replace traditional bottled fermentation. At present, there have been studies on using polyethylene film bags as fermentation containers for Oat Silage [6] and Cacao [7].

Natural fermentation (fermentation carried out at normal temperature without inoculation) is adopted in the fermentation of traditional Apo pickle. Therefore, the fermentation speed varies greatly under different temperature conditions, and the fermentation process is often determined through detection. However, techniques like GC-MS require expensive equipment and skilled technicians [8] and require hours or even days to analyze samples. CSA is a method that can classify target analytes [9,10]. This method has fast reaction speed and is simple to operate. It uses the idea that volatile organic compounds (VOC) cross-react with chemically reactive materials to cause color variations that can be used to detect sample status [11]. CSA technology is currently being utilized to identify the fermentation process of several foods that ferment, such as apple vinegar [12] and black tea [13], and it can also detect the fermentation process of Apo pickle [14]. By improving the indicator points in the array, the detection ability of CSA can be improved.

In this study, examining the diversity and flavor of bacteria in the fermentation process of Apo pickle can provide a theoretical basis for determining the dominant bacteria, controlling the fermentation process, and improving the quality. The feasibility of using vacuum bags instead of plastic bottles as Apo pickle fermentation containers was tested by comparing the chemical indexes and bacterial diversity in the fermentation process of VBA and TBA. In addition, a CSA was developed to monitor the traditional fermentation process of Apo pickle.

## 2. Materials and Methods

### 2.1. Preparation of Sample

*B. napus* was acquired from Huarun Supermarket in Wuxi. After it was washed with running water, it was baked at 50 °C for 1 h and then pickled with 8% salt at 15 °C for 3.5 days. After the pickled *B. napus* was dehydrated, part of it was packed into plastic bottles and compacted, weighing 130 ± 5 g per bottle. After water extrusion, the remaining *B. napus* was packed into a smooth vacuum bag (20 cm × 30 cm, made of polyethylene) and then sealed by using vacuum packing equipment (DZQ-400; Shanghai Youxi Machinery Equipment Co., Ltd., Shanghai, China) (130 ± 5 g per bag). The bottled and bagged pickles were incubated at 20 °C, and three groups of samples were randomly taken for analysis every five days to ensure the repeatability of the experiment.

Experimental-related reagents, including the indicator (Nile red, cresol red, methyl red, bromophenol blue, methyl red sodium salt, phenol red), ethanol, NaOH, phenol-phthalein, sodium tetraborate decahydrate, Zinc acetate dihydrate, potassium hexacyanoferrate (II) trihydrate, Sodium nitrite, Sulfanilic acid, N-1-Naphthylethylenediamine Dihydrochloride, t-butylammonium hydroxide (TBAH), 5,5′-dithiol-bis (2-nitrobenzoic acid)(DTNB), 2,4-Dinitrophenylhydrazine(DNPH), agar, and cyclohexanone were purchased from Macklin Biochemical Technology Co., Ltd. Man, Rogosa, Sharpe (MRS) medium was purchased from Haibo Biotechnology Co., Ltd. (Qingdao, Shandong, China). Polyvinyl fluoride (PVDF) membranes were purchased from BIOS Harper Bio-technology Co., Ltd. (Shanghai, China). Polyethylene vacuum packaging bags were purchased from CNON Packaging Products Co., Ltd. (Shenzhen, China). Other reagents were at least analytical grade, except for cyclohexanone which was GC standard.

### 2.2. Biochemical Index Analysis

The pH and total acid (TTA) levels were determined using the Chinese Industry Standard SB/T 10213-1994 [15] and the Chinese National Standard GB12456-2021 [16]. A 25 g sample (accurate to 0.01 g) was transferred into a triangular flask, 150 mL water, boiled for 30 min, and then the volume was adjusted to 250 mL after filtration. The pH was determined with the help of a pH meter (Lei Ci PHS-3C; Shanghai Instrument Co., Ltd., Shanghai, China), and the TTA was detected by titration with NaOH and converted to grams of lactic acid. N-(1-naphthyl) ethylenediamine dihydrochloride spectrophotometry was used to determine the nitrite concentration of Apo pickle [17].

The total number of LAB colonies during the fermentation of Apo pickle was measured according to the Chinese National Standard GB4789.35-2016 method [18]. A 25 g sample was weighed and homogenized with 225 g 8% NaCl, then gradient diluted with 0.8%Nacl according to the ratio of 1: 10. A total of 2–3 suitable dilutions were selected, 1 mL sample solution was sucked into a Petri dish, and then MRS agar medium cooled to 48 °C was poured into the Petri dish. All plates and the anaerobic culture were repeated thrice at 37 °C for 72 h. The solvents and consumables used in the experiment had to be sterilized, and all operations in LAB testing had to be conducted under aseptic conditions.

### 2.3. Determination of VOC

The method of Huang et al. [19] was referred to and modified appropriately to analyze VOC in the sample. A total of 5 g chopped samples were put into a 20 mL headspace vial and bathed at 60 °C for 10 min. The SPME fiber DVB/CAR/PDMS was exposed in the sample headspace for 30 min before being removed and introduced into the heated GC injection port at 250 °C for 5 min for subsequent analysis. The samples of VBA and TBA on the 35th day of fermentation were semi-quantitatively analyzed, and the internal standard was 5 μL 0.494 mg/mL cyclohexanone methanol solution.

Gas chromatographic conditions: DB-WAX (30 m × 0.25 mm × 0.25 μm) chromatographic column, inlet temperature of 250 °C, carrier gas speed at 1 mL/min, and column temperature at 40 °C for 3 min, which was then increased to 230 °C at 10 °C/min for 6 min.

Mass spectrometric conditions: LECOEI ion source, 210 °C ion source temperature, one mA emission current, 70 eV electron energy, 250 °C transmission tube temperature, full-scan mode, and 33–400 *m*/*z* scan range. 

### 2.4. DNA Extraction and Sequencing of Apo Pickle

HiPure Soil DNA Kit (Mgbio Biology Science and Technology Co., Ltd., Shanghai, China) was used to extract the DNA of the bacterial population from the samples, and the DNA concentration was detected using the Qubit^®^ dsDNA HS Assay Kit (Vazyme biology science and technology co., ltd, Nanjing, China). The bacterial community’s 16S rRNA genes V3–V4 area was generated using primers. The precise 16S rDNA primer sequence was downstream primer 5′-GGACTAC-NVGGGTWTCTAATCC-3 and upstream primer 5′-CCTACGGRRBGCASCAGKVRVGAAT-3. Appendix A provides information on the PCR amplification conditions. Using electrophoresis on a 1.5% agarose gel, the DNA’s integrity was determined. After DNA libraries were mixed, double-ended sequencing was performed based on the Illumina Miseq (Illumina, San Diego, CA, USA) PE 300 platform. The reads obtained by double-ended sequencing were spliced, and short sequences, ambiguous sequences, chimeric sequences, and other sequencing errors were removed. After quality filtering, operational taxonomic units (OTUs), which may be compared to Silva (https://www.arb-silva.de/ (accessed on 27 December 2022), were obtained by clustering the useful sequences using VSEARCH (1.9.6) at a similarity threshold above 97%. A representative sequence was chosen for each OTU for the purpose of obtaining information about the species classification that corresponds to that OTU. The representative sequence was then annotated using the RDP classifier Bayesian algorithm, and statistics were calculated at the classification levels of Kingdom, Phylum, Class, Order, Family, Genus, and Species, respectively. This process was entrusted to Azenta Biological Technology Co., Ltd. (Suzhou, China).

### 2.5. Diversity Analysis

Qiime (1.9.1) examined the diversity index (Ace, Shannon, Simpson, and Chao1) based on the analytical findings provided by OTU. The intergroup with a linear discriminant analysis (LDA) score greater than 3 was examined using the linear discriminant analysis effect size (LEfSe).

### 2.6. Electronic-Nose Detection

We adopted the method by Gao [20]. GC E-nose Heracles II (Alpha MOS, Toulouse, France) was used to analyze pickle. We put 5 g samples into a headspace bottle and prepared four parallel samples for each sample. The vials were incubated at 45 °C for 25 min at 500 rpm. The injector temperature was 200 °C, and the injected volume was 2500 μL.

### 2.7. Preparation of CSA

A total of 10 mg of the indicator was mixed with 5 mL anhydrous ethanol, and ultrasonication was performed for 15 min to obtain a 2 mg/ mL indicator solution. Slight pH changes can have a large impact on the indicator color when the pH of the pH indicator is close to its color transition point. The indicator’s sensitivity may be increased by adjusting the pH using an acid or base close to the point of discoloration [21]. Methyl red and cresol red were added to 0.8 mol/L TBAH according to 3 μL/ mL and fully mixed to prepare acid-bas sensitive indicators. In addition, according to Liu’s method [11], we also prepared indicators that can improve the detection sensitivity of carbonyl and mercapto groups. Specifically, indicators (Nile red, bromophenol blue, and Bromocresol purple) were mixed with 2, 4-Dinitropropylhydrazone (DNPH) and PEG-400 in the ratio of 1: 3: 3 (aldehyde-sensitive indicators) and indicators (methyl red, phenol red and Methyl Red Sodium Salt) are mixed with 5,5′-dithiol-bis (2-nitrobenzoic acid; DTNB), PEG-400, and 0.1 moL/L NaOH in the ratio of 1: 3: 3: 1 (sulfur-sensitive indicators).

Appendix A is 9 kinds of gas-sensitive indicators used. A total of 2 μL is sucked and printed on a PVDF film to form a 3 × 3 sensor array (Appendix A).

### 2.8. CSA Information Collection

The original image information and final image information (pickle react with CSA for 30 min) of the prepared CSA were obtained by using an HP Laserjet M1005MFP flatbed scanner (Hewlett-Packard, Shanghai, China). To obtain a different image, we computed the average RGB (red, green, and blue) values of 100 pixels close to each chosen point, as well as the absolute values of RGB change values of the indicators prior to and following the reaction (the formula is presented below). Each sample rinse was repeated 12 times.
(1)$$∆R=|dR_{a}-dR_{b}|$$
(2)$$∆G=|dG_{a}-dG_{b}|$$
(3)$$∆B=|dB_{a}-dB_{b}|$$
a means after, b means before, and Δ*R*, Δ*B*, and Δ*G* mean color differences.

### 2.9. CSA Data Analysis Method

LDA was used to distinguish the pickle at different fermentation times, and partial least-squares regression (PLSR) was used to predict the biochemical properties of the samples based on CSA to judge the correlation between CSA data and chemical indicators.

### 2.10. Statistical Analysis

With the help of Origin Software 2023 (OriginLab Corporation, Northampton, MA, USA), charts and a heat map of Spearman’s rank correlation were produced. Principal component analysis (PCA) and discriminant factor analysis (DFA) were realized by multivariate statistical analysis using the electronic nose’s AlphaSoft V2021. Using IBM SPSS Statistics 25 (SPSS Inc., Chicago, IL, USA), LDA was carried out. The Unscrambler X 10.4 (Camo Corp., Oslo, Norway) was used for PLSR to determine the correlation between CSA and biochemical indicators. Redundancy analysis (RDA) was performed using Canoco 5 (Wageningen UR, Wageningen, The Netherlands).

## 3. Results and Discussion

### 3.1. Changes of Biochemical Indexes

PH and TTA are two main parameters of pickle fermentation [22]. When pH drops to about 4, and TTA > 7 g/kg, Apo pickle is mature [14]. The PH of VBA and TBA decreased steadily throughout the fermentation, reaching about 4 on the 30th day (Figure 1A).

TTA (Figure 1B) of TBA and VBA exceeded 7 g/kg after 15 d, and the fermentation speed of VBA (that is, pH decreased speed and TTA increased speed) was faster than that of TBA within 15 d. Nitrite in VBA and TBA decreased rapidly within 5 d and then stabilized, and the highest content appeared at 0 d, which was 2.44 mg/kg (Figure 1C). This was because LAB gradually evolved into dominant flora along with the fermentation of pickled vegetables [23]. Studies have shown that a variety of LAB have the ability to degrade nitrite [24]. During the fermentation process of common pickled vegetables, the nitrite content increased sharply in a short time, and the nitrite peak appeared [25,26]. In this study, the environment was clean, the salt concentration was high, and miscellaneous bacteria did not grow vigorously. The LAB quickly became the dominant bacteria, and no nitrite peak appeared.

The LAB (Figure 1D) content of VBA is higher than that of TBA in the whole fermentation process, which may be because vacuum packaging can provide a better anaerobic environment for Apo pickle, which makes the LAB in VBA occupy a dominant position quickly, and inhibits some aerobic microorganisms. LAB may ferment inorganic acids from raw materials (such as glucose and fructose) [27], give pickles a distinctive flavor, and encourage the lowering of pH and rising of TTA content, which causes VBA to ferment more quickly than TBA in the early stages of fermentation.

### 3.2. VOC Analysis of Apo Pickle

During the fermentation of VBA and TBA, a total of 642 VOC were detected by GC- -MS, which were divided into nine categories (The change of content ratio of each category during the fermentation process is shown in the Figure 2A), including 114 main VOC (esters (19), acids (18), phenols (11), aldehydes (18), ketones (6), alcohols (29), sulfides (5), isothiocyanates (8), and nitriles (2)). The content and standard deviation of individual substances are shown in Appendix A. Isothiocyanates are the main odorants with pungent and distinctive odor in fresh *B. napu* [28]. From 0 d to 35 d, the proportion of isothiocyanates in VBA and TBA decreased from 12.51% to 2.19% and 0.85% (Figure 2A), respectively, mainly including the decrease in the proportion of 2-isothiocyanate-Butane, 2-Phenylethyl isothiocyanate, 1-Butene-4-isothiocyanate and n-Heptyl isothiocyanate (Appendix A), which helped to make the aroma milder. Similar results could be obtained for other fermented vegetables [29]. It was shown that isothiocyanates were converted to nitriles during fermentation [1]. Sourness is a characteristic flavor of fermented vegetables. Throughout the fermentation process of VBA and TBA, the proportion of acids and phenols increased steadily (Figure 2A). Acetic acid was the main important VOC in mature Apo pickle, and its content in VBA and TBA fermentation increased from 0.27% (0d) to 18.06% (35 d) and 20.71% (35 d), respectively. In addition, trans-3-Hexenoic acid, tetradecanoic acid, octadecanoic acid, 4-ethylphenol, and maltol, which accounted for a large proportion at the conclusion of fermentation, also contributed to the unique flavor of mature Apo pickle (Appendix A). Aldehydes are chemically active, and most of them are intermediate products. Thus, the content of aldehydes in the Apo pickle showed an increase and then a decrease (14.38–17.44–8.08% in VBA, 14.38–18.58–6.56% in TBA) (Figure 2A). According to PCA (Figure 2B), the two kinds of Apo pickle obviously overlap, which can confirm the types and changing trends of VOC in TBA were similar to those of VBA. In the mature stage of Apo pickle (35 d), except for some VOC which exist in both VBA and TBA and have similar proportions, such as acetic acid (20.17%, 18.06%), 4-ethylphenol (7.61%, 7.44%), 3-hexen-1-ol (3.36%, 4.14%), and ethanol (3.38%, 3.68%), which contributed sour/ pungent, pungent, bell pepper/ grass, alcohol/ floral odors, respectively, there are still many different flavor substances between VBA and TBA, which makes the overall flavor different. Figure 2C is the result of a semi-quantitative analysis of Apo pickle at the mature stage (35 d). It can be concluded that VBA has more volatile flavor substances than TBA, so it has a richer flavor.

### 3.3. Diversity of Bacterial Community during Pickle Fermentation

The prokaryote of pickled vegetables’ 16S rDNA V4-V5 region was amplified and sequenced using primers. The basic OTU and species richness information are shown in Appendix A. The length of PE reads obtained from this test is between 78,530 and 315,623, and the main length of these pruning sequences is 400–420 bp. After quality filtering, the effective sequence length after chimera sequence removal is 54,537–260,150. Using the RDP classifier to analyze the representative sequences of OTU, we could cluster 81–149 unique OTUs belonging to bacteria.

According to the analysis, the rarefaction curves of samples (Figure 3A) reached a plateau in the last period. ACE, Chao1, Simpson, and Shannon indexes (Appendix A) were used to illustrate the richness and uniformity of TBA and VBA during fermentation.

### 3.4. Changes in Bacterial Flora during Apo Pickle Fermentation

The natural fermentation process of Apo pickle is dependent on microorganisms found naturally in *B.napus*, and research into the variations in the bacterial population throughout the fermentation phase of Apo pickle can give a theoretical foundation for further inoculation fermentation of Apo pickle. We studied the bacterial communities in different stages of the VBA and TBA fermentation process and drew the relative abundance map (Figure 3B,C). At the phylum level, the sum of the relative abundance ratios of *Firmicutes* and *Proteobacteria* was above 97% in the whole process, and the content of *Firmicutes* kept rising, becoming the dominant phylum (Figure 3B). This result was the same as that of some traditional fermented products in China, such as pickled radish [30], and fermented sausage [31]. The main genera in the fermentation process of VBA and TBA included *Lactobacillus*, *Weissella*, f__*Enterobacteriaceae*_Unclassified, *Exiguobacterium*, *Aerococcus*, *Pseudomonas*, and f__*Erwiniaceae*_Unclassified. LAB is a key player in the fermentation of Apo pickle [22]. In the initial stage of fermentation, *Weissella* was the LAB dominant genus, and it was heterotypic fermented LAB, which can use the phosphoketolase pathway to ferment carbohydrates and produce ethanoic acid, CO_2_, ethanol, and lactic acid, thus effectively improving the flavor [32]. At 5 d, the relative abundance of *Weissella* was highest in VBA and TBA, reaching 45.53% and 50.67%, respectively. The relative abundance of *Weissella* fell gradually as lactic acid accumulated and acidity increased, reaching 10.19% and 19.42% at 35 d, respectively. *Lactobacillus*, a homofermentative LAB with great acid resistance and stringent anaerobicity, gradually became the dominating genus in the middle and late phases of fermentation. Pyruvate produced during this fermentation was converted into lactic acid by lactate dehydrogenase (LDH), which made the pH decrease. The relative abundance of *Lactobacillus* in VBA reached 78.45%, while that in TBA reached 58.51% at 35 d. In the whole fermentation process, the relative abundance of LAB in VBA > TBA, which may be because the vacuum bag used in the experiment has better gas barrier performance than plastic bottles and can better ensure an anaerobic environment. This needs further exploration. The f_*Enterobacteriaceae*_Unclassified and *Pseudomonas* were the non-LAB dominant genera in samples, with high relative abundance in the early stage; their growth was inhibited by the increase of *Lactobacillus* in the fermentation process, which was similar to the research results from He et al. [33].

The above conclusions show that the dominant genera of VBA and TBA are the same in the fermentation process, but the content of LAB, which plays an important role in fermentation, is different. The LEfSe analysis of VBA during the fermentation is shown in Figure 4. The dominant genera were *Lactococcus*, *Staphylococcus*, *Klebsiella*, and *Weissella* at 5 d, *Aerococcus* at 10 d, and *Lactobacillus* at 35 d.

### 3.5. Correlation among Biochemical Properties, VOC, and Bacterial Community of VBA

Figure 5 is a heatmap of Spearman’s rank correlation (Figure 5A) and RDA (Figure 5B) between biochemical properties, VOC, and bacterial community of VBA. *Lactobacillus* was negatively correlated with *Weissella*, f_*Enterbacteriaceae*_unclassified, *Exiguobacterium*, *Pseudomonas*, and f_*Erwiniaceae*_Unclassified. These findings revealed that *Lactobacillus* emerging in the middle and late stages of fermentation inhibited other genera more effectively than *Weissella* and f_*Enterbacteriaceae*_Unclassified in the early stages of fermentation. *Lactobacillus* displayed a negative association with pH value and a positive correlation with other dominating genera, showing that *Lactobacillus* can create acid to reduce pH and decompose nitrite. Huang et al. also found similar results [23]. The contents of acids and phenols had a positive correlation with that of *Lactobacillus* and a significant negative correlation with other dominant genera (except *Aerococcus*), and other VOC had little correlation with the dominant genera (Figure 5A, B). As shown in Figure 5B, 0d was located at the positive end of shaft-1 and shaft-2, while 5d (Figure 5B) was located at the positive end of shaft-1 and the negative end of shaft-2, which were significantly different from those at other times. The samples after 10 d were linearly arranged, showing that the variety of the bacterial population in each stage of the samples was relatively distinct, and that it was a gradually changing process.

### 3.6. E-Nose Analysis of VBA

Heracles NEO Electronic Nose combines gas chromatography with traditional electronic noses while possessing the qualitative analysis ability for the gas phase and discrimination ability of the traditional sensor, which can separate and identify volatile gas substances in samples [34], as well as group pickle samples by some chemometric analysis, such as PCA and DFA [35]. PCA of a rapid gas-phase electronic nose is shown in Figure 6A. The discrimination index of Apo pickle (Figure 6A) was 83, the contribution rate of variance of PC-1 was 58.74%, and that of PC-2 was 24.84%. The components of the VOC varied significantly during each measurement; the Apo pickle at different fermentation times could be identified. On the basis of PCA, DFA made the difference between the data in the same group as small as possible and enlarged the difference between the data in different groups to further confirm the PCA results (Figure 6B).

### 3.7. Detection of CSA in the VBA Fermentation Process

Both GC-MS and an electronic nose can qualitatively and quantitatively identify the volatile gases to distinguish and determine the pickled-vegetable fermentation process, but their operation is very tedious. CSA imitates the olfactory system of mammals through the cross-reaction of a variety of dyes, which can quickly and intuitively distinguish different samples according to color changes. It can also quantify the measurement results combined with digital image-processing technology [36]. According to the results of GC-MS, the acid content in VBA increased with the fermentation, accompanied by the transformation of sulfur, aldehydes, and other substances. In this study, a simple CSA was prepared to react with VBA samples at different fermentation stages. Compared with CSA prepared by Wang et al. [14], We added aldehyde-sensitive indicator points to improve the detection sensitivity of CSA. The CSA color-difference maps of VBA during fermentation are shown in Figure 7A. The acid content increased during the VBA fermentation process, and the brightness of the CSA color-difference maps of indicator points 2 and 3 sensitive to pH change gradually increased, while the color of the color-difference maps of points 1, 5, and 8 sensitive to aldehydes changed at different fermentation times. Compared with the different images obtained by Wang et al. [14], the visual difference between the difference image obtained in different fermentation states in this study is more obvious.

Studies have shown that olfactory visual sensor technology combined with appropriate chemometric methods can provide qualitative and quantitative analysis [37]. LDA data of CSA grouped by fermentation days are shown in Figure 7B; the sample classification accuracy of cross-validation reached 93.8% (Appendix A). If the CSA data is grouped into three stages according to the information in Figure 7B (the three stages are 0 d (early fermentation), 5 d and 10 d (middle fermentation), and 15–35 d (late fermentation), and are then analyzed by LDA (Figure 7C), then the accuracy of sample classification of cross-validation can reach 100% (Appendix A), indicating that it is feasible to distinguish VBA in different fermentation stages using simple CSA.

### 3.8. PLSR Analysis

The PLSR analysis was used in this study to build a quantitative prediction model to predict the biochemical properties of VBA, and its performance evaluation indexes included calibration correlation coefficient RC2, prediction correlation coefficient Rp2, root mean square error of calibration (RMSEC), and root mean square error of prediction (RMSEP). RMSEC is used to measure the deviation between the predicted value and the measured value, and RMSEP can be used to evaluate the predictive ability of the model for unknown samples. Generally, the smaller the RMSEC and RMSEP are, the better the prediction ability of the model [38]. PLSR analysis of CSA against pH, TTA, and nitrite (Figure 7D–F) was performed. The correlation coefficients of Rc^2^ were 0.836, 0.837, and 0.873, and RMSEC were 0.247, 0.0897, and 0.273. Rp^2^ were 0.803, 0.802, and 0.823, and RMSEP were 0.276, 0.0994, and 0.327, indicating that the data acquired by the olfactory visual sensor had a certain correlation with the changes of VBA chemical properties and CSA results had good prediction ability for the chemical properties of VBA.

## 4. Conclusions

In this study, we attempted to use the vacuum bag as the fermentation container of *B. napus*. The changing trend of the measured basic biochemical indexes was similar to that of TBA, which shows that this idea is feasible. Further analysis showed that the number of LAB in the VBA was higher than that in TBA during the whole fermentation process. The pH decrease and TTA content growth rates in the early fermentation stage were faster than those in the bottle, most likely because vacuuming in the bag can offer a better environment for LAB fermentation. The combination of CSA and chemical analysis methods, such as LDA, could accurately distinguish the different fermentation stages of VBA and play a role in monitoring the fermentation process. 

In conclusion, low-pressure fermentation using a vacuum bag could replace atmospheric-pressure fermentation using plastic bottles, and CSA could detect the fermentation process. Future research should examine how to improve the packaging form of the bag so that CSA can be placed in the fermented bag but not in contact with the vegetable to realize real-time determination of the fermentation process.

## Figures and Tables

**Figure 1 foods-12-03573-f001:**
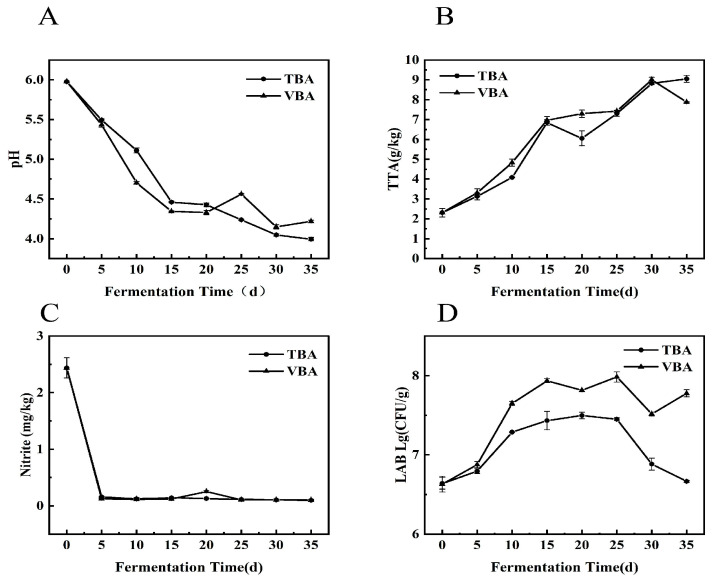
Biochemical properties of VBA and TBA during fermentation. (**A**) pH; (**B**) TTA; (**C**) nitrite; (**D**) LAB.

**Figure 2 foods-12-03573-f002:**
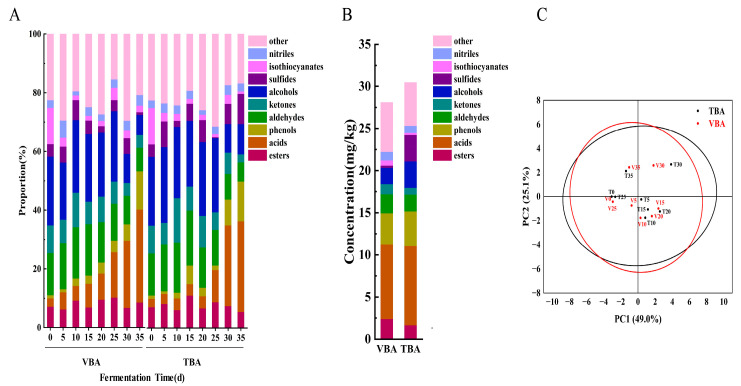
Variation of VOC during Apo pickle fermentation. (**A**) The proportion of VOC during the whole fermentation process of VBA and TBA; (**B**) The concentration of VOC in VBA and TBA on 35d; (**C**) The PCA of the VOC of VBA and TBA fermentation process. V represented VBA; T represented TBA; Numbers (0, 5,10, 15, 20, 25, 30, 35) represent fermentation time.

**Figure 3 foods-12-03573-f003:**
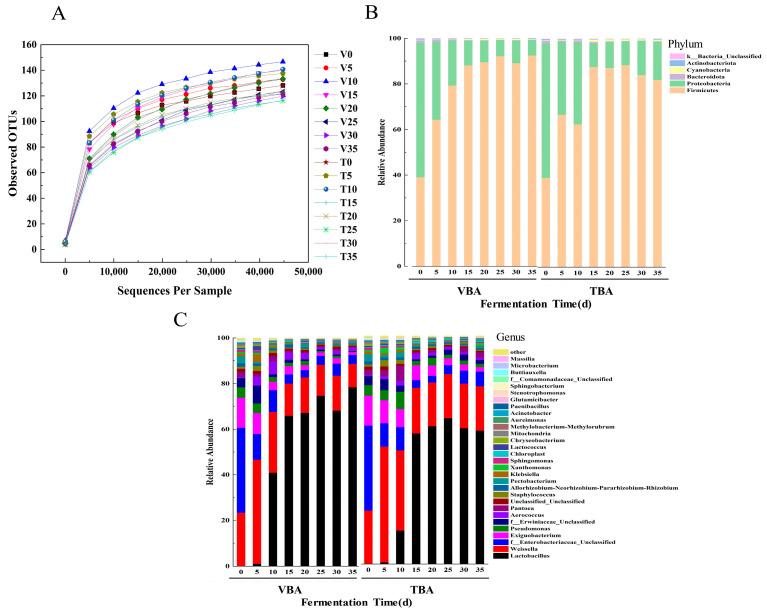
The Illumina HiSeq sequencing of Apo pickles. (**A**) Bacterial community OTU rarefaction curves; (**B**) the bacterial community of VBA and TBA at the phylum level; (**C**) the bacterial community of VBA and TBA at the genus level.

**Figure 4 foods-12-03573-f004:**
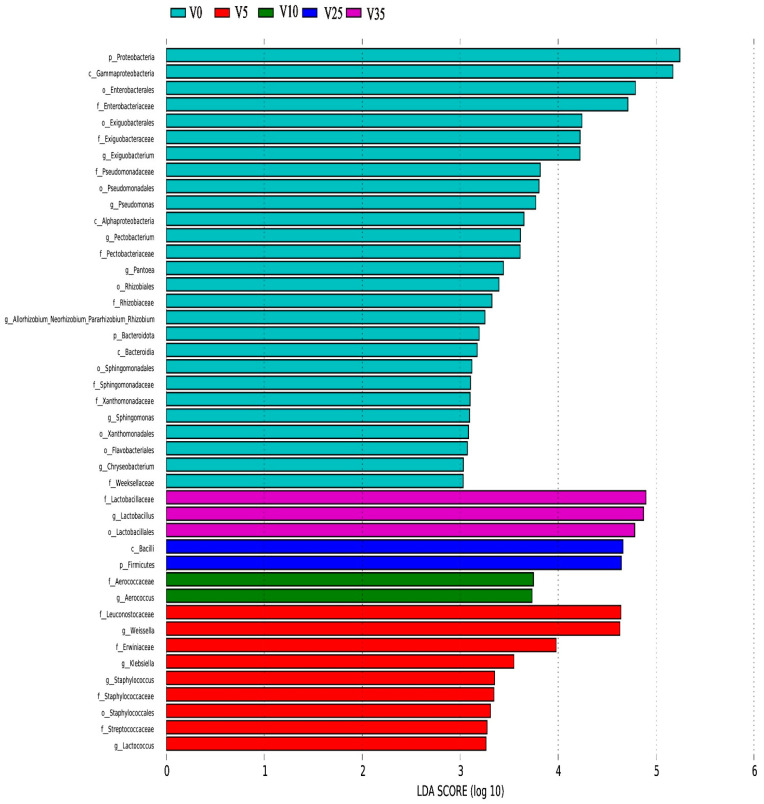
VBA LEfSe analysis.

**Figure 5 foods-12-03573-f005:**
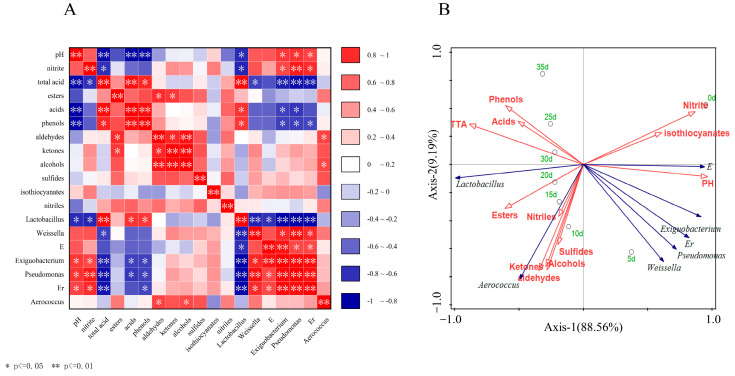
(**A**) A heatmap of Spearman rank correlation. Blue denotes a negative association for their content, and red denotes a positive correlation; (**B**) RDA between biochemical properties, VOC, and bacterial community in the fermentation. Process of VBA. * *p* ≤ 0.05, ** *p* ≤ 0.01. E, Er represented f_*Enterobacteriaceae*_Unclassified, f_*Erwiniaceae*_Unclassified. Green fonts represent time, blue arrows represent different kinds of bacteria and microorganisms, and red arrows and fonts represent biochemical indicators and VOC.

**Figure 6 foods-12-03573-f006:**
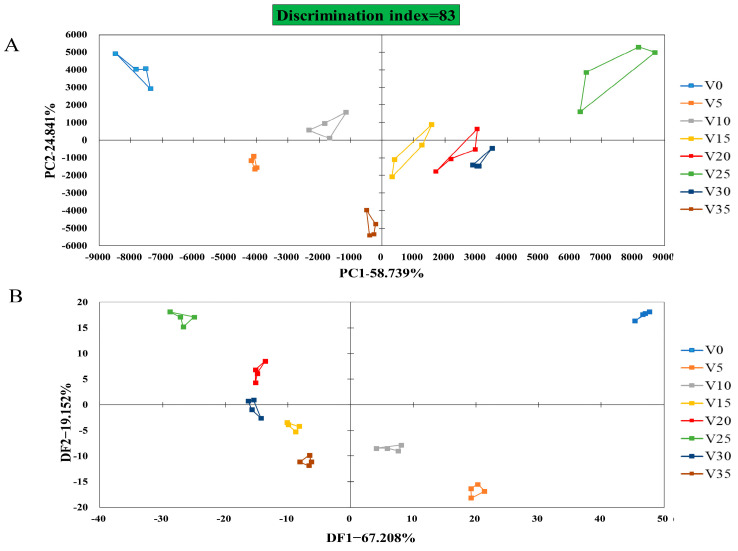
Electronic nose analysis of VBA. (**A**) PCA; (**B**) DFA. V represented VBA; Numbers (0, 5, 10, 15, 20, 25, 30, 35) represent fermentation time.

**Figure 7 foods-12-03573-f007:**
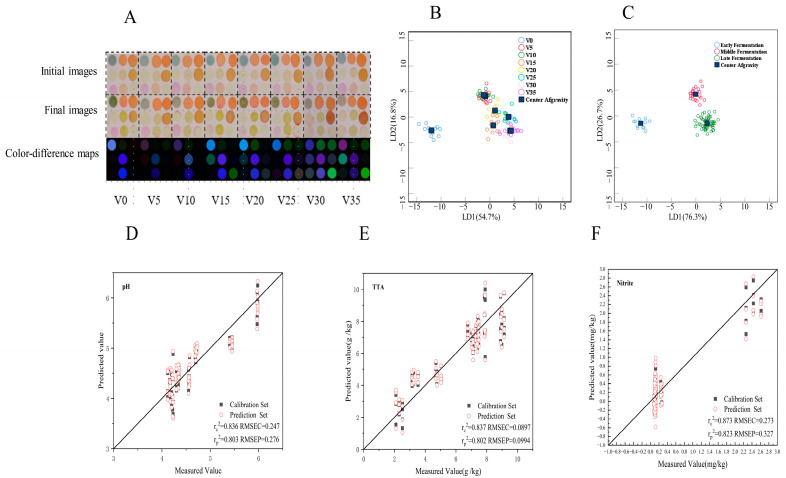
Data analysis of VBA by CSA detection. (**A**) Images of CSA for VBA at various stages of fermentation; (**B**) LDA data of CSA grouped by fermentation days; (**C**) LDA data of CSA grouped by different fermentation stages; (**D**) PLSR outcome plot of pH for VBA based on the CSA; (**E**) PLSR outcome plot of TTA for VBA based on the CSA; (**F**) PLSR outcome plot of nitrite for VBA based on the CSA. V represented VBA; Numbers (0, 5, 10, 15, 20, 25, 30, 35) represent fermentation time.

## Data Availability

The datasets generated for this study are available on request to the corresponding author.

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
