# Peer review of "Study on the Characteristics of Vacuum-Bagged Fermentation of Apo Pickle and Visualization Array Analysis of the Fermentation Process"

_foods, 2023, doi:10.3390/foods12193573_

Round 1
Reviewer 1 Report
General/major comments
In this paper, the authors present a study aiming to analyze the chages of Apo pickle's flavor and microbial diversity during two different fermentation processes (bottled and bagged). This is an interesting study although the added value linked to carrying out vacuum-bagged fermentation rather than bottled fermentation is not clearly established.
The introduction needs to be reworked; it does not really introduce the work but rather appears as a catalog of the techniques used. For example, the fermentation aspect is rarely discussed in the introduction.
The results and discussion section does not call for any particular remarks, the authors compare the two fermentation processes in terms of physicochemical characteristics, lactic bacterial flora and content of volatile organic compounds. Even if this remains quite descriptive, this work provides arguments in favor of fermentation in vacuum bags.
The manuscript requires careful rereading to correct typing errors or typos throughout it.
The manuscript could be suitable for publication in Foods after considering these minor suggestions for improvement.
Specific comments
Be sure to write in italics all genus or species names of microorganisms throughout the manuscript.
L11: The first sentence of the summary must be rewritten (please include a verb), as it currently stands it is incomprehensible.
L12: Change "Brassica napus" to "Brassica napus"
L13: The sentence "the results show…" should be corrected and/or rewritten
Have the manuscript proofread by an English speaker
Reviewer 2 Report
In the present study, the authors have evaluated and compared the fermentation profile of Apo pickle between two types of fermentation procedures using colorimetric sensor array (CSA) method. The aim of the present study seems to be of interest.
Although there are some typographical errors in the submitted manuscript, the present study seems to be well designed and conducted, the submitted manuscript almost satisfies the criteria for publication, however, there are some concerns in the manuscript for acceptance at present form as follows:
Major
At line 66–67, the authors have mentioned that no studies on applying CSA method to the fermentation process of fermented vegetables, however, the authors have previously published the article regarding the detection of fermentation process of Apo pickle using CSA method (Wang et al., Foods 2022, 11, 3577). The previous paper reported the fermentation profile of Apo pickle using additionally inoculated LAB strains. The authors should cite the previous work with appropriate discussion on common and different observations. In addition, the CSA method used in the present work seems to be improved from the previous one, thus the description on the differences between them will make the present study more valuable.
Minor
There are some typographical errors in the manuscript.
e.g. At 112, kimchi samples?
Genus name should be written in italics.
(In Minor comments)
There are some typographical errors in the manuscript.
e.g. At 112, kimchi samples?
Genus name should be written in italics.
Reviewer 3 Report
Here are some suggestions to improve the research article titled "Study on the characteristics of vacuum-bagged fermentation of Apo pickle and visualization array analysis of the fermentation process" in the following sections:
Plagiarism Index: 22% (Reduce it up to permissible limit)
Abstract:
- Clarity: Be explicit about the primary objective of the research right from the beginning. For instance: "This study aims to compare...".
- Conciseness: Some sentences can be merged for brevity. For example, "The fermentation speed of VBA was faster than that of TBA in the early stage, and the lactic acid bacteria content of VBA was higher than that of TBA in the whole fermentation process, but their flavors were similar."
- Categorization: Break the abstract into clearly defined subsections, such as 'Objective', 'Methods', 'Results', and 'Conclusion' to enhance clarity.
Introduction:
- Historical Background: Provide more context on the historical significance and use of Apo pickle.
- Relevance: Why is the study of Apo pickle's fermentation process significant? Perhaps highlight its relevance in culinary, health, or economic contexts.
- Problem Statement: Clearly state the issues with the current fermentation process, and what challenges or gaps this research aims to address.
- Objectives: Clearly define the primary and secondary objectives of the research.
Materials and Methods:
- Details: Some subsections, like DNA extraction and sequencing, could benefit from more detailed explanations for readers unfamiliar with the specifics.
- Standardization: Ensure all units, equipment, and procedures are standardized, especially when referencing specific protocols or methods.
- Replicability: Highlight if any part of the study, like the sampling, was done in duplicates or triplicates, ensuring replicability of the study.
- Safety & Ethics: Mention any safety precautions taken during the experimental process, especially given the biological nature of the study.
- Organize Methods Sequentially: Try to align the methods in the order they were conducted in the experiment. This makes the process clearer to the reader.
- Citations: Ensure that all referenced methods are correctly cited, and all citations are listed in the reference section.
Results and Discussion:
- In-depth Microbial Analysis: While the analysis touched upon the microbial communities, deeper insights into the microbial dynamics might provide better clues as to why LAB dominates in the VBA fermentation compared to TBA.
- External Variables: Factors such as temperature, external contaminants, and light exposure can influence fermentation processes. Considering the variables that could affect fermentation in a vacuum bag compared to a bottle might be insightful.
- Sensory Analysis: It would be beneficial to conduct a sensory analysis comparing the flavor, aroma, and texture of products from both fermentations. This would give a practical perspective on the acceptability of the final product to consumers.
- Economic Analysis: From a commercial perspective, understanding the cost implications of transitioning from bottle to vacuum-bag fermentation would be crucial. This could encompass production, transportation, and storage costs.
- Environmental Impact: Analyzing the environmental footprint of using vacuum bags over plastic bottles would be insightful, especially in the current global scenario where sustainability is paramount.
- Safety Measures: The safety of using vacuum bags for long-term fermentation needs to be ensured. This includes understanding if any harmful chemicals leach from the bags into the pickles over time.
- Shelf-life Study: The shelf-life of the fermented product in vacuum bags vs. bottles could be an important parameter to consider, especially for commercial applications.
- Improved CSA Deployment: As mentioned, improvements can be made in how the CSA is deployed to ensure it doesn't come into direct contact with the fermenting product.
- Feedback Loop: Regular feedback from both a control group (traditional fermentation method users) and an experimental group (vacuum-bag fermentation method users) can be incorporated into the research for continuous improvement.
- Expand to Other Ferments: It would be interesting to test if the vacuum-bag method is efficient for other types of ferments, such as sauerkraut, kimchi, or kombucha.
General Comments:
- Consistency: Make sure there's consistency in naming the samples throughout the paper. For instance, ensure that 'VBA' is used uniformly and not interchanged with other terms.
- Formatting: Ensure proper formatting in terms of line numbers, font, spacing, and indents.
- Language: Refine the language for clarity and simplicity. Consider proofreading to rectify any grammatical errors.
By addressing these areas, the research can provide a more holistic view of the advantages and potential limitations of using vacuum bags over traditional fermentation containers.
Minor editing of English language required
Round 2
Reviewer 2 Report
The revised manuscript may be improved appropriately.
Reviewer 3 Report
Accept in present form